# Colon Cancer: From Epidemiology to Prevention

**DOI:** 10.3390/metabo12060499

**Published:** 2022-05-30

**Authors:** Kyriaki Katsaounou, Elpiniki Nicolaou, Paris Vogazianos, Cameron Brown, Marios Stavrou, Savvas Teloni, Pantelis Hatzis, Agapios Agapiou, Elisavet Fragkou, Georgios Tsiaoussis, George Potamitis, Apostolos Zaravinos, Chrysafis Andreou, Athos Antoniades, Christos Shiammas, Yiorgos Apidianakis

**Affiliations:** 1Department of Biological Sciences, University of Cyprus, Nicosia 2109, Cyprus; katsaounou.kyriaki@ucy.ac.cy (K.K.); teloni.savvas@ucy.ac.cy (S.T.); 2AVVA Pharmaceuticals Ltd., Limassol 4001, Cyprus; e.nikolaou@avvapharma.com (E.N.); c.shammas@avvapharma.com (C.S.); 3Stremble Ventures Ltd., Limassol 4042, Cyprus; paris.vogazianos@stremble.com (P.V.); cameron.brown@stremble.com (C.B.); athos.antoniades@stremble.com (A.A.); 4Department of Electrical and Computer Engineering, University of Cyprus, Nicosia 2109, Cyprus; stavrou.g.marios@ucy.ac.cy (M.S.); andreou.chrysafis@ucy.ac.cy (C.A.); 5Institute for Fundamental Biomedical Research, Biomedical Sciences Research Center Alexander Fleming, Vari 16672, Greece; hatzis@fleming.gr; 6Department of Chemistry, University of Cyprus, Nicosia 2109, Cyprus; agapiou.agapios@ucy.ac.cy; 7Nicosia General Hospital, Nicosia 2029, Cyprus; elislida@cytanet.com.cy (E.F.); tsiaoussisgeorgios@yahoo.com (G.T.); 8Potamitis Gastroenterology and Nutrition, Nicosia 1082, Cyprus; g.potamitis@logos.cy.net; 9Department of Life Sciences, European University Cyprus, Nicosia 1516, Cyprus; a.zaravinos@euc.ac.cy; 10Basic and Translational Cancer Research Center, Nicosia 1516, Cyprus

**Keywords:** colorectal cancer, epidemiology, prevention, risk factors, intestinal microbiota, inter-individual diversity, intra-individual variation, regenerative inflammation, multi-omics, gut-on-chip

## Abstract

Colorectal cancer (CRC) is one of the most prevalent cancers affecting humans, with a complex genetic and environmental aetiology. Unlike cancers with known environmental, heritable, or sex-linked causes, sporadic CRC is hard to foresee and has no molecular biomarkers of risk in clinical use. One in twenty CRC cases presents with an established heritable component. The remaining cases are sporadic and associated with partially obscure genetic, epigenetic, regenerative, microbiological, dietary, and lifestyle factors. To tackle this complexity, we should improve the practice of colonoscopy, which is recommended uniformly beyond a certain age, to include an assessment of biomarkers indicative of individual CRC risk. Ideally, such biomarkers will be causal to the disease and potentially modifiable upon dietary or therapeutic interventions. Multi-omics analysis, including transcriptional, epigenetic as well as metagenomic, and metabolomic profiles, are urgently required to provide data for risk analyses. The aim of this article is to provide a perspective on the multifactorial derailment of homeostasis leading to the initiation of CRC, which may be explored via multi-omics and Gut-on-Chip analysis to identify much-needed predictive biomarkers.

## 1. Introduction

The Cypriot and international market lacks a validated testing tool able to estimate an individual’s risk for colorectal neoplasia. The products approved by FDA for CRC screening, such as Faecal Occult Blood Test (FOBT), Guaiac Faecal Occult Blood Test (GFOBT), and Faecal Immunochemical Test (FIT), can produce false positive or false negative results because many cancers do not result in bleeding, and bleeding does not specifically herald cancer. Cologuard, a non-invasive stool DNA CRC screening test detecting blood and abnormal cells shed into the colon, claims to detect 92% of stage 1–4 colorectal cancers. However, this method shares the limitations of the aforementioned faecal tests, and it is not reliable for people with a personal or family history of colon cancer, as well as those diagnosed with Inflammatory Bowel Disease or cancer syndromes, such as Familial Adenomatous Polyposis. Most importantly, none of the existing tests can estimate an individual risk for CRC based on molecular alterations of the normal-appearing colonic mucosa before morphological changes occur. Nor do they exploit the transcriptional and microbiota signatures that could be predictive of neoplasia. A multi-omics approach may instead provide realistic prospects for early molecular biomarker analysis, an adjunct tool to conventional colonoscopy. Biomarkers indicating susceptibility to CRC could be used for cost-effective screening and medical assessment. Some may be modifiable and reversible with probiotic and prebiotic administration or via dietary changes able to remodel the gut environment and the mucosal response to external stimuli.

Here we focus on CRC epidemiology, analyzing the role of environment, genetic and molecular pathways, intestinal microbiota, and metabolites leading to or facilitating the development of CRC. We pinpoint factors based on their potential as proxies for CRC risk or causal biomarkers that provide opportunities for targeted preventive interventions. To capture the big picture of environmental and molecular factors, biological modalities are categorized and cross-linked in an overview of lifestyle, genomic, epigenomic, transcriptomic, proteomic, metabolomic, and metagenomic interactions (Figure 1). Multi-omic platforms and gut-on-chip technologies are considered key tools for a better understanding of the underlying cross-linked mechanisms leading to CRC.

## 2. Colon Cancer Epidemiology: The Role of Environment

Epidemiology aims to explain the distribution and evolution of diseases within and among populations [1], tracing individuals in location and time to ultimately determine the modi operandi of the disease [2]. During the last few decades, CRC remains the third most common malignancy in men and the second most common in women worldwide in terms of incidence. It is also the third deadliest cancer in the United States, where, despite the decrease in deaths from 49,190 in 2016 to 37,930 in 2020, the morbidity rate is estimated to increase 10-fold by 2035 [3,4]. Notwithstanding the geographic variation in factors such as life expectancy, socio-economic profiles (Human Development Index, HDI scores), diet, and lifestyle, the prospect of increasing morbidity is globally apparent [3,5].

The typical European and North American lifestyle choices of red meat and alcohol consumption, sedentary lifestyles, low fibre diets, smoking, and obesity have led to consistently high rates of CRC in many countries. Asian societies (India, Japan, Saudi Arabia) are catching up in recent years in CRC incidence by gradually adopting the ‘Western models of life’ [6]. While better diagnosis and preventive medical practices lower the risk, by 2035, global CRC incidence is expected to increase by 60%—with 2.2 million cases and 1.1 million deaths projected yearly—due to the anticipated economic growth in low-to-medium Human Development Index (HDI) countries [7,8]. Not only the citizens of developed nations but also immigrants from low-to-medium HDI nations are at increased risk. Combined with the prospect of increases in carcinogen emissions and exposures linked to climate change, CRC risk may increase even more [9].

### 2.1. Modifiable Risk Factors with a Clear Environmental Component

#### 2.1.1. Inactivity, Sedentary Lifestyles and Obesity

Sedentary lifestyles have become more prevalent during the last fifty years, primarily in Europe and North America. The overall sitting time associated with desk jobs is consistently increasing in high-income countries [10]. Physically active individuals present with a 25% lower risk of developing colon and rectal tumours, as exercise benefits the cardiovascular and gastrointestinal systems and increases blood flow as well as metabolic rates. This, in turn, decreases both waist circumferences and body mass indexes (BMIs), which are somatic features linked to CRC [11]. Circumstantial factors may also exacerbate detrimental lifestyles. For example, 70% of young people decreased their physical activity and increased their sitting time due to the global measures against the SARS-CoV-2 pandemic [12]. Moreover, physical activity and obesity are affected by environmental factors, such as air and water pollution and access to green areas, sports facilities, and parks. The worse the environmental quality provided by the state, the higher the obesity prevalence linked to the lack of physical exercise [13].

According to the World Cancer Research Fund (WCRF) International, obesity increases the risk for CRC incidence by 50% in men and 20% in women. Subcutaneous fat normally serves as an energy deposit within the body [14], but as adiposity increases, fat penetrates deeper into the visceral zone, accumulating ectopically in the liver, heart, skeletal muscles, pancreas, and gut, inducing metabolic imbalances, such as hepatic steatosis, fatty liver disease and other chronic diseases including cancer [15,16]. The accumulation of visceral fat deep in the liver, pancreas, and intestine interferes with organ homeostasis, stimulating the secretion of hormones and pro-inflammatory cytokines, such as TNF-a, leptin, IL-1β, IL-6, IL-7, and IL-8 from adipocytes, which in turn promote oxidative stress and immunosuppression, and induce chronic low-level inflammation via higher C-reactive protein and serum amyloid A in the blood [11,16,17]. Similarly, excess body fat is associated with higher levels of sugar, insulin growth factor I, insulin, and insulin-like growth factor-binding proteins 1 and 6 in the blood [16,17]. Inflammatory and growth factors, in turn, may induce metabolic and endocrine disorders and oncogenesis. However, studies of leptin-deficient mice point to multiple conflicting factors regarding the role of obesity, suggesting that obese individuals are not necessarily prone to metabolic disorders and systemic low-grade inflammation. Obese people having healthy cholesterol levels and normal blood pressure are not more prone to developing obesity-linked type II diabetes or cancer [15].

#### 2.1.2. Tobacco Consumption

In 2009, smoking was established as the main preventable cause of mortality due to any type of cancer. CRC is strongly associated with heavy and long-term tobacco smoking. Smoking heavily (>40 cigarettes/day) for over 30 years increases the risk of CRC incidence by 40% and doubles CRC mortality compared to non-smokers [18].

Interestingly, heavy smoking is associated with increased rectal and proximal but decreased distal colon cancer incidence. Tobacco smoke comprises hundreds of carcinogenic compounds causing genetic and epigenetic abnormalities, such as mutations in BRAF signalling components, high micro-satellite instability (MSI-high), and the CpG island methylator phenotype (CIMP-high). Tobacco use may influence these pathways to induce serrated polyps in the rectum and proximal colon [11,19]. While DNA methylation patterns change dramatically in smokers compared to non-smokers or former smokers and may contribute significantly to CRC initiation, the effects of smoking become progressively reversible upon quitting [20].

#### 2.1.3. Overconsumption of Red and Processed Meat

Case-control and cohort studies point to an association between red and processed meat consumption and an increased risk of polyps, mostly adenomas, in the colon and rectum [21]. A high intake of red and processed meat increases the incidence of colorectal cancer malignancies by 20–30%, most likely because the overcooking of red meat at high temperatures generates carcinogens [22].

Red meat refers to the meat derived from the muscles of domesticated animals (beef, pork, lamb) and game animals. Processed meat is modified by smoking, curing, salting, and fermenting to enhance its flavour or to improve its preservation. Studies in mice established the mutagenic role of N-nitroso compounds (NOCs), heterocyclic amines (HCAs), polycyclic aromatic hydrocarbons (PAHs), heme-iron, and secondary bile acids (SBAs) derived from over-cooked red meat [23]. Nitrates and amines or amides derived from processed and overcooked meat generate NOCs, such as DNA-damaging nitrosamines and nitrosamides [24]. HCAs and PAHs are formed during long, high-heat, or open-air red meat cooking and are potent genotoxic compounds able to induce carcinogenesis [25].

Heme, a porphyrin ring carrying a charged iron atom, is found in very high amounts in red meat muscle. Heme iron induces reactive oxygen species (ROS), promoting DNA mutagenesis. Moreover, heme-iron molecules can be metabolized to generate NOCs. The lower concentration of heme in white meat and fish (almost 10-fold lower) may explain the non-carcinogenic status of these products and that high fish intake protects against CRC [26].

Cholic and chenodeoxycholic acids are the primary bile acids synthesized in the liver. They are typically conjugated to glycine or taurine before their secretion in the intestinal lumen and their subsequent conversion by colonic bacteria to SBAs. The latter, in turn, may promote epithelial cell membrane perturbation, metabolic and endoplasmic reticulum stress, oxidative and mitotic stress, DNA repair inhibition, and cell apoptosis or necrosis. The inflammation induced by necrotic cells and their by-products may further promote tumorigenesis via epithelial cell hyperproliferation and hyperplasia [23].

#### 2.1.4. Alcohol Consumption

A global meta-analysis of 14 cohort studies indicates that even light daily alcohol consumption elevates CRC risk significantly [27]. Modest and heavy social drinkers have a 20% and 40% increased risk for colorectal cancer formation compared to non-drinkers, respectively. Men usually consume more alcohol than women, in agreement with a stronger association between alcohol consumption and increased risk for CRC incidence in males [17]. Ethanol is metabolized by bacterial alcohol dehydrogenase to toxic and reactive acetaldehyde, which accumulates in the colon, presumably due to the low aldehyde dehydrogenase activity of the colonic mucosa. Acetaldehyde, in turn, can promote DNA methylation and cause damage [28]. Of note, daily alcohol consumption of >24.6 g ethanol increases CRC risk by about 30%, promoting CRC tumours that are usually BRAF and KRAS wild type, MSI-stable (MSS), and CpG island methylator low/negative (CIMP-low/neg) [29].

#### 2.1.5. Dietary Fibre and Whole Grains

Dietary fibre and whole grains may prevent the formation of colon polyps and CRC, according to WCRF. A possible prophylactic mechanism involving high fibre intake is mediated by the binding of fibre to primary and secondary bile acids and carcinogens [30,31,32]. Moreover, whole grain fibre increases bowel movements, reducing faecal transit time and bowel exposure to pro-inflammatory factors and carcinogens [31,32]. Short Chain Fatty Acids (SCFAs) produced upon the intake of whole grains and bacterial fermentation within the large intestine, such as acetate, propionate, and butyrate, reduce primary to secondary bile acid harmful by-products by decreasing the luminal pH. Butyrate may also induce the apoptosis of cancer cells and colonic cancer cell arrest, along with normal colonic mucosa regeneration. A possible mechanism involves NF-κΒ inhibition by butyrate and a concomitant reduction in the cytokines TNF-α and IL-6 [31,33]. The strong association of whole grain and fibre intake with a decreased risk of CRC prompted the WCRF to recommend the daily consumption of at least 20–30 g of dietary fibre.

#### 2.1.6. Dairy Products and Dietary Supplements

Dairy products derived from domesticated animals and milk, in particular, may lower colon tumour incidence, mainly because of their high concentration of calcium. Calcium may share common mechanisms with dietary fibre because it can bind to SBAs and fatty acids, eliminating their inflammatory and carcinogenic potential [22]. Calcium may impede cell division and decrease DNA mutations and damage by promoting cell apoptosis and differentiation [34].

Vitamin D3, the main facilitator of calcium and phosphorus absorption in the body, lowers CRC risk, presumably by regulating colonic epithelial cell cycle genes towards normal differentiation and apoptosis. It also has anti-inflammatory effects, enhances immune responses, and intensifies angiogenesis. Vitamin D has been shown to reduce colon polyp recurrence, while consistent intake of Vitamin D is strongly associated with better overall survival of colorectal cancer patients by regulating the WNT signalling pathway that promotes oncogenesis [35].

#### 2.1.7. Age

CRC incidence increases with age. Approximately 90% of new cases involve people over the age of 50 years, whereas persons over the age of 65 are three times more likely to develop CRC [36]. There is an alarming increase in early-onset CRC manifested in individuals between the ages of 20–49 in Europe and North America, presumably prompted by rapidly evolving westernized lifestyles [37,38].

## 3. The Molecular Epidemiology of Colon Cancer: The Role of Genetic Variability

Although more than 90% of CRC cases are sporadic, one-fourth of which occur within families (familial CRC), the remaining are inherited in a Mendelian way, predisposing individuals born with single-gene mutations [19,39].

Epigenetic abnormalities observed in CRC include the CpG island methylator phenotype (CIMP), promoted by hypermethylation of the CpG-rich promoter regions of tumour suppressor genes (TSGs) [40]. TSG mutations may also arise from chromosomal abnormalities (CIN), microsatellite instability (MSI), and the mutation or inhibition of the DNA mismatch repair system (MMR), which may in turn cause MSI. CIN is caused by numerical aneuploidy, polyploidy, or structural chromosomal disorders, while the MSI phenotype involves changes in the number of repetitive tandem DNA sequences (satellite DNA) near the ends of chromosomes due to mutated *MMR* genes [41,42].

MSI phenotypes are classified into three categories with respect to the frequency of the phenotype within the genome: MSI-low (MSI-L), MSI-high (MSI-H), and MSI-stable (MSS). Of note, MSI-H DNA aberrations are found in nearly 15% of all CRC cases, including Hereditary Non-Polyposis Colorectal Cancer (HNPCC or Lynch Syndrome).

There is a strong correlation between the CIMP and the MSI phenotype, as the hypermethylation of CpG islands leads to *MMR* gene mutations and eventually to an MSI-H (hypermutable) phenotype [43]. Clinical studies involving over 2000 participants identified sporadic CRC tumours exhibiting all three molecular states (CIN, CIMP, MSI-H). Approximately 80% of these tumours presented with the CIN phenotype, 20% with the CIMP phenotype, and 10–15% were MSI-H [44].

### 3.1. Risk Factors with a Clear Genetic Component

#### 3.1.1. Hereditary Colorectal Cancer Syndromes

Approximately 5% of all CRC cases result from inherited syndromes, most commonly Lynch Syndrome (or HNPCC) and Familial Adenomatous Polyposis (FAP). Both disorders are characterized by autosomal dominant inheritance [19,45]. The following heritable syndromes collectively account for approximately 2% of CRC incidence: Peutz–Jeghers syndrome (PJS), MUTYH-associated polyposis (MAP), Turkot syndrome (a sub-type of FAP), Juvenile polyposis syndrome (JPS), PTEN hamartoma tumours syndrome (PHTS), Mixed polyposis syndrome, and Serrated polyposis syndrome [45,46].

*Lynch Syndrome (HNPCC).* Lynch syndrome accounts for 2–4% of all CRC cases. MSI-H and MMR deficiency characterizes approximately 90% of HNPCC tumours. The MSI-H status in HNPCC is induced by germline DNA *MMR* gene mutations, predominantly of the *hMLH1*, *hMSH2*, *hPMS2*, and *hMLH6* genes. Notably, *MLH1* methylation serves as a CIMP biomarker and an early marker of this syndrome [40]. Patients with HNPCC tend to present with CRC malignancies at a young age. Moreover, there is an increased likelihood of right-sided (proximal colon) CRC development. HNPCC patients have a very high probability of CRC development, estimated at 50% [19,45].

*Familial Adenomatous Polyposis (FAP).* FAP is the second most common hereditary CRC syndrome and accounts for approximately 1% of all CRC cases. All FAP patients are afflicted by CRC by their forties but can be sub-grouped as A-FAP (attenuated, patients with fewer polyps) and Turkot syndrome (rarer syndrome with higher numbers of adenomatous polyps and higher CRC incidence) [45]. FAP is driven by germline APC mutations on chromosome 5q21. Hundreds to thousands of adenomatous polyps develop in FAP patients (mostly in the distal colon) due to *APC* gene mutation and a concomitant β-catenin-mediated aberrant induction of cell growth [47]. The identification of APC mutations and the establishment of *APC* as a gatekeeper against tumour formation paved the way for understanding the mutational sequences leading to CRC [48,49].

#### 3.1.2. Inflammatory Bowel Disease (IBD)

IBD is the third most common genetic risk factor after HNPCC and FAP. It refers to a group of chronic inflammation conditions of the gastrointestinal (GI) tract, including Crohn’s Disease (CD) and Ulcerative Colitis (UC) [11,22]. CD appears as GI wall-layer-penetrating inflammation involving various areas of the digestive tract, while in UC, inflammation and tissue damage are restricted to the colonic epithelium [11,50]. Chronic inflammation is a hallmark of carcinogenesis because it elicits growth factors (EGF, VEGF), cytokines, chemokines, and pro-angiogenic factors, facilitating cancer initiation and progression [51]. The relative risk of CRC development in IBD patients is estimated to be increased by 4 to 20-fold [36].

The deregulation of various signalling pathways appears causal for IBD predisposition. Genes in the following pathways are linked to the risk of IBD: epithelial cell junction assembly, innate immune recognition of microbes, GPCRs and immune defence, anti-inflammatory (interleukin-10) signalling, Th17-cell differentiation, B-cell activation, and IgA antibody production [50].

#### 3.1.3. Personal History of Colonic Adenomas

Colonic and rectal polyps can be classified into two categories: neoplastic or dysplastic polyps (tubular and villous adenomas) that predispose an individual to CRC, and hyperplastic polyps, which are benign, non-inflammatory, and do not predispose CRC lesion formation [52]. Even though only 5% of dysplastic adenomas become malignant, they give rise to nearly 95% of sporadic CRC cases [53]. The neoplastic to malignant transformation is a slow process requiring the acquisition of a handful of key mutations and may take 5 to 10 years [54]. Consequently, the increased risk of CRC development in individuals with a personal history of colonic polyps is reduced through the detection and complete removal of those lesions [36,52].

#### 3.1.4. Comorbidities: History of Diabetes and Other Diseases

Diabetes mellitus is a metabolic disorder that may induce inflammation and increase CRC incidence by 20–30% [22,55]. The *PTPN2* gene, which encodes the T-cell protein tyrosine phosphatase (TCPTP), may partially explain the increased risk for CRC neoplasia due to diabetes [56,57]. For example, *PTPN2* regulates IL-1β production, which in turn down-regulates pro-inflammatory responses in ulcerative colitis (UC) but simultaneously induces CRC development [55,56,57].

Cystic fibrosis, a syndrome with genetic aetiology, as well as cholecystectomy and abdominal radiotherapy, may also elevate the risk for CRC, but the mechanisms remain unclear [11].

#### 3.1.5. Sex

Males are more prone to sporadic CRC development than females due to a complex interaction between human physiology and socio-cultural trends and factors, such as male–female differences in diet, alcohol consumption, heavy smoking, and supplement intake [58]. However, right-sided colon cancer is more frequent in women, who also exhibit a lower 5-year-survival rate compared to men over 65 years of age [11,59,60].

The differences in CRC incidence in men have led many researchers to implicate sex hormones in CRC development. Sex hormones and other endogenous metabolites appear to promote some cancers, including CRC [61]. For example, women under postmenopausal hormone therapy are likely to present with a lower risk of CRC development [62]. Moreover, high expression of testosterone and decreased oestradiol to testosterone ratios associate with lower CRC incidence [61]. Nevertheless, CRC predisposition through oestrogen, oestradiol, and testosterone depends on environmental risk factors, such as dietary patterns, supplement intake, and obesity [63].

Epidemiological data regarding the east-Mediterranean island of Cyprus point to CRC as the third most common cancer type in both men and women, with the rate of new cases in 2020 being approximately two-fold higher in men (12.9%) compared to women (7.2%) [64]. This sex-linked difference, while extreme in this country, agrees with the global epidemiological picture.

#### 3.1.6. Self-Reported Race/Ethnicity

The most up-to-date data from the American Cancer Society show that CRC rates display significant discrepancies between different ethnicities [11,45]. Across the American continent, the United States and Brazil show the highest rates of CRC incidence, with ratios of 6.8% and 9.3%, respectively [3]. The CRC incidence ratio is 20% higher in African Americans, especially for early-onset CRC incidence, compared to any other ethnic/racial group in the USA. This disproportional incidence is directly associated with socio-economic profiles, which influence health care access and life quality in general [45,65]. Of note, Asians and Pacific Islanders have the lowest rate of new cases when compared with all races and ethnicities worldwide [66,67]. Colorectal cancer statistics worldwide reported largely elevated CRC incidences in Asian populations. CRC has been reported to be most prevalent in Japanese individuals (14.1%), followed by Russians (13.1%) and Chinese (12.2%). Although India is still among the top 10 countries with the highest incidence worldwide, its CRC incidence is markedly lower (4.9%). CRC incidences among the top four European countries with the highest incidence (Italy, UK, France, and Germany) vary between 11.7% and 9.2%. These data appear aligned to the HDI of the countries [3,68].

### 3.2. Genome Wide Association Studies

Over the last two decades, millions of SNPs have been identified through Genome-Wide Association Studies (GWAS). Meta-analysis studies, mostly of European and Asian populations, have linked hundreds of SNPs to elevated CRC risks. A 2015 analysis by Al-Tassan and colleagues mentions 20 genetic loci accounting for 8% of the familial CRC risk, including the *MYC*, *BMP4*, *BMP3*, *POLD3*, *TERC*, *CDKN1A*, and *SHROOM2* genes [69]. Other GWAS studies in Poland and Japan link the SNP rs10935945 in the long non-coding RNA gene *LINCO2006* at the 3q25.2 locus, near the *FOXF1* gene, with an elevated risk of CRC development and metastasis [70]. Similarly, SNP rs6065668 on 20q13.12, located in the promoter of the *TOX2* gene, has a variant conferring higher *TOX2* expression, which is linked to higher rates of advanced-stage CRC in Japanese people [71].

## 4. Intestinal Microbiota Deregulation as a Risk Factor for CRC

Trillions of microorganisms reside in the human GI tract. Prokaryotes (archaea and bacteria), unicellular eukaryotes (fungi), and viruses, including bacteriophages, form relatively stable microbial communities interacting with host metabolism and immunity factors [72]. The intestinal ecosystem protects the host from pathogen colonization in accordance with the host’s diet and antibiotic treatments [73,74,75]. Commensal bacteria of the GI tract are held in check by antimicrobial peptides secreted in the intestinal crypts, while luminal antigens help maintain intestinal barrier integrity via immune cell stimulation and mucus production [76,77,78].

### 4.1. Intestinal Pathogens Linked to CRC

Several bacterial species have been shown to play a role in CRC development (Table 1). The oral anaerobe *Fusobacterium nucleatum* can travel through the bloodstream into colorectal adenomas, promoting CRC initiation and progression. Patient-derived xenografts in murine models point to the advancement of colorectal tumorigenesis in the presence of *F. nucleatum* [78,79,80]. Moreover, *F. nucleatum*-positive CRC tumours show resistance to the chemotherapeutic drug oxaliplatin because *F. nucleatum* promotes the survival and proliferation of CRC cells via Toll-like receptor 4 (TLR-4)-mediated induction of autophagy in those cells [80,81,82].

*Pks+ Escherichia coli* and *Enterotoxigenic Bacteroides fragilis* (ETBF) are also found in adenomas and CRC tumours in great abundance. These microbes, together with *F. nucleatum*, constitute a well-established triad of bacterial strains able to promote CRC. ETBF induces acute inflammation along the GI tract, generating microbial communities, termed biofilms, layering and encircling adenomas and CRC tumours [81,82,87]. Tumour formation in *Apc*^Min/+^ mice can be induced by ETBF, whose secreted toxin (BFT) may bind colonic epithelial cells causing proteolysis of the tumour suppressor E-cadherin. Moreover, the stimulation of the IL-6/STAT3 inflammatory pathway and the subsequent induction of T helper type 17 T cell responses (Th17 immuno-response) may promote inflammation-induced colonic neoplasia [88,89]. *Pks*+ *E. coli* produces colibactin, a genotoxic compound, which induces DNA damage in the epithelial cells of the colon [81]. AOM/DSS-treated, IL10-deficient mice, can develop invasive carcinomas due to colibactin-mediated DNA damage and epithelial barrier disruption [90]. Similarly, *Campylobacter jejuni* and *Salmonella* can produce DNA damaging toxins and colorectal neoplasia [91].

Beyond the well-established bacterial instigators, *Streptococcus bovis*, *Enterococcus feacalis*, and *Peptostreptococcus anaerobius* may also colonize colorectal adenomas, inducing inflammation and concomitant oxygen radicals in the colonic submucosa [91]. *F. nucleatum* and *P. anaerobius* generate a pro-inflammatory immune microenvironment in *Apc*^Min/+^ mice, leading to the immune cell infiltration of tumours and their progression [92,93]. Moreover, *Parvimonas micra*, *Peptostreptococcus stomatis*, and *Akkermansia muciniphila* may also induce colorectal neoplasia together with *F. nucleatum*, but their role in CRC development and clinical significance needs to be further investigated [94,95].

### 4.2. Human GI Microbiome Inter-Individual Diversity

The infant microbiome starts to be shaped directly after birth via contact with the mother and the feeding processes. Naturally delivered and breastfeeding infants exhibit increased colonization by genera, such as *Bacteroides*, *Bifidobacterium*, *Lactobacillus*, *Streptococcus*, *Staphylococcus*, and *Propionibacterum*, whereas C-section delivered babies and exclusively formula-fed babies typically display a high abundance of *Enterobacteriaceae* and *Clostridium deficile* [96,97,98,99]. When solid food is established, infant gut microflora starts to gradually resemble a state of adult-mature microbiota [100].

Whereas a core microbiome appears to be shared among disparate human individuals [101], one’s state of health is associated with certain metabolomic functions and microbial genes [102,103]. Differences in microbial colonization may lead to chronic inflammation disorders, including IBD, Type I Diabetes, Celiac disease, and neurodegenerative disorders, highlighting the fundamental link between intestinal microbiota composition and long-term health [104].

Thousands of bacterial species shape the microflora of the adult GI tract [105,106]. *Bacteroidetes* and *Firmicutes* dominate the microbiome, amounting to about 90% of the bacterial species residing in the gut, followed by *Actinobacteria* and *Proteobacteria*. *Bacteroidetes* and *Proteobacteria* are involved in the modulation of immunity, carbohydrate metabolism, and defence against pathogens [105,107]. The complex adult ecosystem also includes, at much lower levels, *Fusobacteria*, *Verrucomicrobia*, and *Cyanobacteria* [108].

Inter-individual differences in the ratios of the most abundant phyla correlate with IBD, obesity, neurological disorders, and CRC risk [109,110]. Indicative metagenomic studies have shown a strong association between obesity and a low ratio of *Bacteroidetes* to *Firmicutes* [109]. Additionally, a sex-specific variation in the *Bacteroides* to *Prevotella* genera ratio has been recorded as higher in men, with *Bacteroides thetaiotaomicron* potentially serving as a sex-discriminative faecal biomarker [109,111]. Obese individuals or those inflicted by chronic inflammatory disorders, such as IBD, carry a low overall count of microbial genes compared to the high gene count found in non-obese and healthy-appearing individuals. The former tends to have more *Bacteroidetes* and *Proteobacteria*, while the latter has a higher abundance of *Actinobacteria*, *Verrucomicrobia*, and *Euryarchaeota* [110].

### 4.3. Inflammation and Intestinal Dysbiosis

Microbiome diversity provides the potential for the identification of biomarkers for inflammatory diseases and CRC prevention [112]. Exogenous factors and several medical conditions can cause dysbiosis, altered microbial composition, and pathogenic interactions with the host, promoting chronic low-grade inflammation [102] (Table 1).

A main function of the intestinal mucosal barrier is to separate host-immune cells from the GI microflora and microbial antigens. Tight junctions stabilize intestinal epithelial cells (IECs), forming a monolayer across the site of the gut facing the lumen [91]. Intestinal permeability can be disrupted by epithelial inflammation; therefore, faecal metabolite biomarkers signifying inflammation may be indicative of intestinal barrier breach [113].

## 5. Intestinal Metabolomics as an Emerging Way to Study CRC Prevention

Humans take up dietary nutrients and work along with the gut microbiomes to produce metabolites, many of which may appear in the faecal metabolome. The SCFAs acetate, butyrate, and propionate comprise a triad of health-promoting metabolites appearing in faecal samples. Each one or in combination may play key roles in intestinal homeostasis. Butyrate, for example, is generated by bacterial fermentation of resistant starch and dietary fibre and, to a lesser degree, from proteins. It is an energy provider for colonocytes and a regulator of epithelial cell mitosis, pro-inflammatory cytokines IL-6 and IL-12p40, and nitric oxide production [114,115]. Butyrate may also reduce pathogen proliferation and concomitant DNA damage via the reduction of faecal pH [116].

Human gut microbes collectively bear nearly 22 million genes, whereas the whole human genome contains only approximately 23 thousand. Heterogeneity between individuals is more prominent in gut microbiome genes than that in identifiable microbiome species [117]. Thus, in addition to the interplay between operational taxonomic units (OTUs) of the gut microbiome, microbial gene content and metabolite secretion need to be considered. Luminal metabolites are closely associated with host physiology, as well as a long list of pathological states, from auto-immune diseases and allergies to obesity and several cancer types, including CRC [118]. Metagenomic sequencing of human faecal samples may identify bacterial species and genes varying among individuals, which may be functionally linked to variable metabolites. Thus, individual faecal metabolite profiles can be cross-linked with the corresponding microbial species and gene profiles, as well as with environmental factors, such as diet, towards an assessment of CRC risk [119].

Alternative high-throughput metabolomic approaches have been utilized to catalogue metabolites in human faeces and intestinal biopsies. To identify, quantify, and authenticate unknown metabolomic derivatives of clinical and biochemical significance, Gas Chromatography (GC), Liquid Chromatography (LC), Ultra-High-Performance LC (UHP-LC) coupled with Mass Spectrometry (MS) and Nuclear Magnetic Resonance (NMR) have been implemented [120,121,122,123]. Hundreds of molecules can be detected using full-scan data acquisition with non-targeted single or tandem MS. To obtain accurate and high-quality data, it is necessary to evaluate primary analysis with alternative methods [120]. NMR allows the accurate identification and quantification of compounds found in high abundance in biofluids, but molecules in very low concentrations are not detectable. On the other hand, the high-resolution power (Quadrupole-Time of Flight, Q-TOF) of GC-MS and LC-MS platforms allows the separation of low-in-abundance and structurally similar volatile metabolites. Moreover, LC-MS and UHPLC-MS can detect a wider range of polar and non-polar metabolites. Taking into consideration the great sensitivity for extremely low molecular weight and the mM to nM concentration of metabolites in human samples, GC-MS, LC-MS, and UPLC-MS are preferred for chemical identification and quantification [123,124,125,126,127]. However, Mass Spectrometry Imaging (MSI) techniques coupled to others, such as Ion Mobility Spectrometry (IMS) [128], and MS techniques coupled to others, such as Capillary Electrophoresis (CE), exemplify analytical methods suitable for metabolomics [124,129,130]. Basic features of modern metabolomic techniques are comparatively displayed in Table 2.

Many human colon metabolites are detectable, and metabolic pathway analysis is possible via any of the metabolomic methods (Table 3). Volatolomics, a GC-MS based metabolomics approach suitable for faecal and colon biopsy analysis, focuses on volatile compounds, including SCFAs. To become more effective towards biomarker identification, volatolomics need to bypass current limitations, such as chemical complexity, dietary, microbial, and lifestyle as confounding factors, a lack of standardized protocols for sampling, sample storage, and analysis, as well as the number of volunteers required [134,135]. These challenges share commonalities with those of other high-throughput analysis platforms. Thus, a multiplatform (multi-omic) analysis may act synergistically to reduce all types of potential errors and facilitate combinatorial biomarker identification.

## 6. Epigenetics and Site-Specific Microenvironment

### 6.1. Intra-Individual (Regional) Differences in Colon Cancer Risk

The large intestine contains (i) the mucosa and glands, (ii) the connective tissue of submucosa and lamina propria, the vasculature and muscle, and (iii) the neurons. These derive from the embryonic endoderm, mesoderm, and neural crest, respectively. Middle (yolk sac) and terminal (caudal) embryonic tissues give rise to the prospective midgut and hindgut, respectively. The midgut–hindgut division is based on differences in arterial supply: hindgut derivatives are supplied by branches of the superior mesenteric artery and include the proximal colon regions, cecum, ascending and two-thirds of the transverse colon, while hindgut derivatives are supplied by branches of the inferior mesenteric artery and include the distal one-third of the transverse, the descending and sigmoid colon and rectum [138]. Despite the peculiarity of transverse colon division in terms of arterial supply, a pathologically defined distinction in the proximal (right) and distal (left) colon is based on the transverse to the descending colon junction [138,139] (Figure 2). Proximal colon cancers usually exhibit a CIMP-H and MSI-H phenotype and are more often mutated in *BRAF* and *PIK3Ca*, whereas distal colon cancers more often exhibit chromosomal instability (CIN), mutations in *KRAS*, *APC*, and *p53*, and the amplification of *HER1* and *HER2* genes [140,141]. Moreover, proximal dysplasia more often exhibits a deregulation of the HER2 (ErbB2) receptor, MAPK, TGF-beta, and insulin signalling pathways, while in distal malignancies, the activation of EGF and WNT signalling pathways are more common [141]. These regional discrepancies may, in part, stem from developmental and, thus, epigenetic differences between the proximal and distal colon.

Moreover, differences in the mucosal microbiome diversity in bacterial biofilms of the colonic epithelium, as well as the luminal levels of bile acids and their metabolites between the proximal and distal colon, may influence carcinogenesis [142,143]. Strikingly, 89% of malignant tumours of the proximal colon were reported to be colonized by invasive luminal biofilms, compared to only 18% of the distal colon. Bacterial biofilms may induce IL-6 and STAT3 activation accompanied by increased cell proliferation and a reduction in E-Cadherin, leading to the disruption of colonic epithelial cell adhesion. Moreover, the pro-mitosis metabolite, N1-N12-diacetylspermine, may be significantly upregulated in mucosal biofilm sites [144,145].

Thus, invasive colonic mucosal biofilm detection may provide a novel biomarker in terms of prevention and treatment. However, regional disparities need to be considered based on the distinct features of the proximal vs. distal colon (Figure 2). For example, the conjugated derivatives of primary cholic acid, taurocholic acid, and glycocholic acid are more than 10-fold concentrated in the proximal colon. Similarly, the secondary bile acid, deoxycholic acid, a metabolic product of anaerobic intestinal bacteria, is found mostly in aspirates from the cecum rather than from rectum faecal samples. Such intra-intestinal variation in bile acid levels may contribute to differential CRC risk [119].

Regional variation in the colon microbiota and its downstream metabolomic derivatives is evident in normal versus overweight and obese individuals. The ascending colon of obese adults may exhibit higher levels of the lipid metabolite trimethylamine-N-oxide and of primary (chenoxycholate) and secondary (taurodeoxycholate) bile acids, but a lower abundance of certain endocannabinoids [136]. This metabolic profile is linked to the overconsumption of certain foods and is conducive to the induction of colon inflammation [136,146]. Furthermore, there is a relative abundance of *Bacteroides* in the ascending colon and of *Proteobacteria* in the descending colon of normal weight individuals [136,147]. The ascending colon of normal weight adults appears to be enriched in *Ruminiclostridium* spp., *Ruminococcus glavus*, and *Tyzzerella* spp., whereas the descending colon in species of the *Barnesiella*, *Faecalibacterium*, *Parabacteroides*, *Parasutterela*, and *Roseburia* genera. These distinct microbial patterns denote differences in metabolomic pathways and cytochemical procedures in the two regions of the colon [136].

Regional differences pertaining to tumorigenesis are not confined to humans. *Drosophila* and mammals share highly conserved intestinal features in terms of physiology, regeneration, and tumour development [148]. The adult *Drosophila* midgut is regenerated via actively dividing stem cells [149,150] and is divided into ten molecularly and anatomically distinct regions that exhibit differences in their propensity for dysplasia [151,152]. Compartmentalization is established during development and appears to be maintained throughout the adult fly life [152]. Accordingly, the downregulation of the tumour suppressor gene, Notch, induces intestinal stem cell clustering almost exclusively in the posterior midgut [151]. It is, therefore, possible that differential stem cell behaviour along the human large intestine and rectum explains some of the regionality of human colon dysplasia and tumorigenesis (Figure 2).

### 6.2. Stem Cell Divisions, Regenerative Inflammation, Cell Differentiation, and Colon Cancer Risk

Tissues undergoing high rates of stem cell division are more prone to malignant transformation [153]. According to the stem cell division theory of carcinogenesis, mutations may randomly arise as normal stem cells or de-differentiated tissue cells divide and give rise to clones of mutated cells [154,155]. The theory explains why a tissue with a high regeneration capacity, such as the colorectal epithelium, is much more prone to malignant transformation compared to a slowly regenerating tissue, such as the heart muscle. However, the stem cell division rate is insufficient to explain the much higher CRC incidence compared to small intestinal cancer incidence since both tissues are highly regenerative [156]. In other words, the stem cell division rate of a tissue/organ alone is insufficient to explain tumour-predisposition factors since carcinogenesis is a very complex and multifactorial disease. Nonetheless, stem cell division rate may be a universal factor of carcinogenesis, and potentially applicable to CRC prevention. For example, isogenic wild-type *Drosophila* strains exhibit extreme variation in the number of midgut stem cell divisions with or without infection, with certain ‘extreme’ strains being very lowly mitotic, while strains on the other side of the spectrum are hyper-mitotic. Moreover, the high and low mitotic strains cope with enterocyte damage and regeneration in two functionally and consequentially distinct ways: the former resort to increasing cell proliferation, while the latter resort to increasing the growth of differentiated enterocytes. While both adjustments contribute to intestinal host defence against infection, highly mitotic strains are more prone to dysplasia [157]. Such strain-to-strain differences in stem cell behaviour may mirror inter-individual human variation in terms of colonic dysplasia and tumorigenesis.

Mitosis and regenerative inflammation genes can serve as biomarkers of risk because they vary among individuals per intestinal site and potentially over time (Figure 3). Most prominent among the factors able to increase stem cell division rate are inflammation and concomitant regeneration [158,159,160]. An accumulation of cancer driver gene mutations can also be linked to environmental exposures, such as microbial toxins and dietary factors [161]. Moreover, the higher tendency for proto-oncogene mutation in proximal rather than distal colon cancers points to site-specific epigenetic explanations [140,141].

Model organisms may provide clues to this end. *Drosophila* midgut stem cell division rates exhibit strain-to-strain and site-to-site variation and are not fixed in time but rather adjust to stress, diet, and ageing. For example, bacterial pathogen or toxin-fed flies exhibit enterocyte damage and a concomitant increase in midgut stem cell mitosis [159,160,161,162,163,164,165]. Aged female flies produce significantly more progenitor cells at the expense of differentiated enterocytes [166]. Similarly, dietary-restricted flies exhibit reduced stem mitosis and dysplasia upon ageing [167]. Anticipating an analogous adaptation of colonic progenitor cell divisions in humans may provide a chance to shed more light on the stem cell division component that may drive and explain CRC development.

## 7. Discussion

The wide range of factors linked to CRC requires a multifactorial approach in future research projects targeting it. In the last two sections, we consider the multi-omics approaches that help to identify clinically relevant biomarkers and gut-on-chip technologies that may provide an efficient way of experimenting with human gut tissues in a personalized way.

### 7.1. Multi-Omics Analysis for Identifying Biomarkers That Can Be Modified by Pre- and Pro-Biotics towards CRC Prevention

Colonoscopy, currently the most effective CRC prevention method, can detect morphological changes in the intestinal mucosa but is insufficient for the detection of the early molecular alterations predisposing for tumorigenesis and dysplasia. The presence of genetic and epigenetic alterations before the appearance of morphological alterations in the normal-appearing colonic mucosa (NAM) of cancer patients is supported by many studies [144,148,162,168,169]. Moreover, dense bacterial communities in the human colonic mucus layer correlate with CRC incidence [142,170]. Experiments in model organisms show a direct synergism between damaging intestinal bacteria and host genetic predisposition in stem-cell-mediated tumorigenesis [145,171,172]. Attempts to correlate the etiologic factors, including genetic, epigenetic and environmental—together with microbial—factors, that influence the microenvironment towards CRC appearance have been undertaken [167,173,174]. Significant correlations of bacterial genera and biofilms with the expression of epithelial cytokines and metabolites have been discovered [170,175]. However, no current method in clinical use can capture the subclinical variations in gene expression and the regenerative inflammation of the colorectal NAM, which may estimate an individual’s risk for colorectal neoplasia.

However, there is promise in identifying valuable molecular biomarkers of risk via multiplatform approaches. Several metagenomic studies have been combined with metabolomic technologies to molecularly distinguish samples from healthy controls vs. CRC or IBD patients, patients with advanced adenomas, or obese individuals. Such studies can identify metabolites and metabolome variations along the colon, as well as the metabolic pathways being activated in the presence of certain chemicals [83,84,85,136]. A study combining microbiome analysis (16S rRNA sequencing) and UPLC-MS metabolomics of faecal samples from patients with advanced adenomas, CRC patients, and healthy controls reported a tight interaction between differences in faecal cholesteryl esters and sphingolipids and increased levels of species belonging to the *Fusobacterium*, *Parwimonas*, and *Staphylococcus* phyla in CRC patients, while decreased levels for bacteria of the *Lachnospiraceae* family. Interestingly, the genus *Adlecreutzia* was more abundant in advanced adenoma patients than in CRC patients, indicative of an early CRC biomarker [83]. Another study combined 16S microbiome analysis and GC-MS metabolomics of faecal samples from healthy individuals and CRC patients showing a significant abundance of *Actinobacteria* and *Firmicutes* in healthy controls, along with an abundance of *Fusobacteria*, *Lentisphaerae*, and *Proteobacteria* in the faeces of CRC patients. This was correlated with the significantly higher levels of polyamines, amino acids, and urea detected in the faeces of CRC patients via GC-MS. However, another study showed a correlation between gut microbiota and metabolites in CRC patients, suggesting reduced gut-microbial richness as a hallmark of low-grade inflammation predisposing to cancer [85].

Considering the multifactorial nature of CRC aetiology, future experimental designs should integrate all of the available data modalities, focusing not only on main effects through univariate analyses but also on identifying interaction effects between biomarkers across and within the same data modalities (Figure 1). Whole-genome sequencing may be impractical for biomarker identification to begin with, as thousands of healthy and cancer-prone volunteers may need to be sequenced to reach statistical significance. However, when large biobank data become available, GWAS may allow the examination of genetic variation and provide polygenic risk scores [69,70,71], which may correlate with other modalities of intestinal health and disease, such as microbiota composition, metabolic output, and mucosa gene expression profiles. Prioritizing transcriptomics, metagenomics, and metabolomics platforms may provide an easier starting point to identify CRC risk biomarkers. Exploiting artificial intelligence, complex relationships, and the underlying mechanisms of disease may be revealed, from the molecular to the phenotypic.

Over the last two decades, Mass Spectrometry Imaging (MSI) systems have emerged as suitable methods for the quantitative profiling and spatial distribution of metabolites, peptides, and other biomolecules on sectioned tissues [176,177]. Thousands of biomolecules, such as lipids, peptides, metabolites, proteins, and glycans can be spatially depicted on thin tissue sections [131]. Only minimum manipulation of the sample is required, which involves flash-freezing for tissue preservation [131,132,176,177]. Broadly used metabolomic techniques paired with MSI are MALDI, DESI, SIMS, and EASI [131,132]. These technologies are currently established for the quantitative assessment and spatial profiling of tissue sections and tissue-derived biofluids in the clinical diagnosis and prognosis of many cancer types, including CRC [178,179,180]. However, they are yet to be applied as tools for CRC prevention. In the foreseeable future, MSI could be applied along with the other metabolomic methods available (Table 1) on healthy-appearing colonic biopsies to identify, quantify, and spatially depict potential CRC risk biomarkers (Figure 4).

RNA-Seq can detect the sequence and abundance of RNA molecules present at a particular time in a specific cell type, tissue, or organ and may reveal the presence and quantity of each messenger RNA in a biological sample [175]. Transcriptomics and metabolomic analysis of normal-appearing mucosa, in combination, should provide a big picture of the gut response to intestinal microbes and metabolites, paving the way for the identification of modifiable biomarkers for prevention and contributing to the development of personalised medicine. Analysing transcriptomics together with biochemical and environmental factors in colon cancer risk could provide insights into how the intestine interacts with environmental factors and the role of intestinal microbes within the colon (microbiome) prior to disease [181]. Moreover, 16S sequencing (16S-Seq) enables the quantification of bacterial ecology in the gut by amplifying and sequencing well conserved and universally distributed regions of 16S rRNA genes [182]. The use 16S-Seq of colonic biopsies may indicate the phylogenetic diversity present in faeces, which could then be linked to the microbial abundance per colonic mucosa site, as well as the data from the mucosal RNA-Seq and metabolomics platforms.

### 7.2. Gut-on-Chip: Towards Experimentation with Personalized Human Gut Tissues

Host–microbiome interactions and their deregulation can have a pronounced effect on the development of CRC [183]. Ex vivo studies enable experimentation with individual host sample tissues, but 2D static cultures cannot simulate intestinal 3D structures, such as villi, mechanical motion (peristalsis), and fluid flow. In addition, animal models are hampered by interspecies differences in immunity and tissue pathophysiology with humans [184,185,186].

Organ-on-chip systems have thus emerged in the last decade, which are microfluidic cell culture devices combining expertise from engineering and cell biology, producing complex geometries and a continuous flow inside the microchambers where cells are cultured, as shown in Figure 5 [187]. Gut-on-a-chip devices recapitulate the structural and functional characteristics of the gut, such as peristalsis [188,189], a 3D microstructure, and hypoxia [190]. They may also allow a more accurate evaluation of personalized drug, nanotherapeutics, and metabolite responses [191,192], as the constant flow of growth medium in the microfluidic channel allows for the controlled and dynamic distribution of nutrients and the removal of cell waste [193].

While conventional culture methodologies yield unrealistic bacteria growth and culture cell damage, the gut-on-chip enables a more physiological interaction of microbes with the gut epithelium [194]. For example, Shigella infects enterocytes more effectively in a gut-on-chip endowed with a continuous nutrient flow and peristalsis [195]. Furthermore, pathogenic *E. coli* strains can be observed in a gut-on-chip together with barrier epithelia and immune cells and antibiotics or probiotics [196], and intestinal cells co-cultured with *Lactobacillus rhamnosus* (strain GG) alone and together with *Bacteroides caccae* exhibit significantly changed transcription and metabolism [197]. Overcoming the challenges of simulating a true intestinal environment, gut-on-chip systems may push personalized medicine one step forward by allowing experimentation and the predisposition assessment of normal and tumorous samples from each individual.

**Figure 5 metabolites-12-00499-f005:**
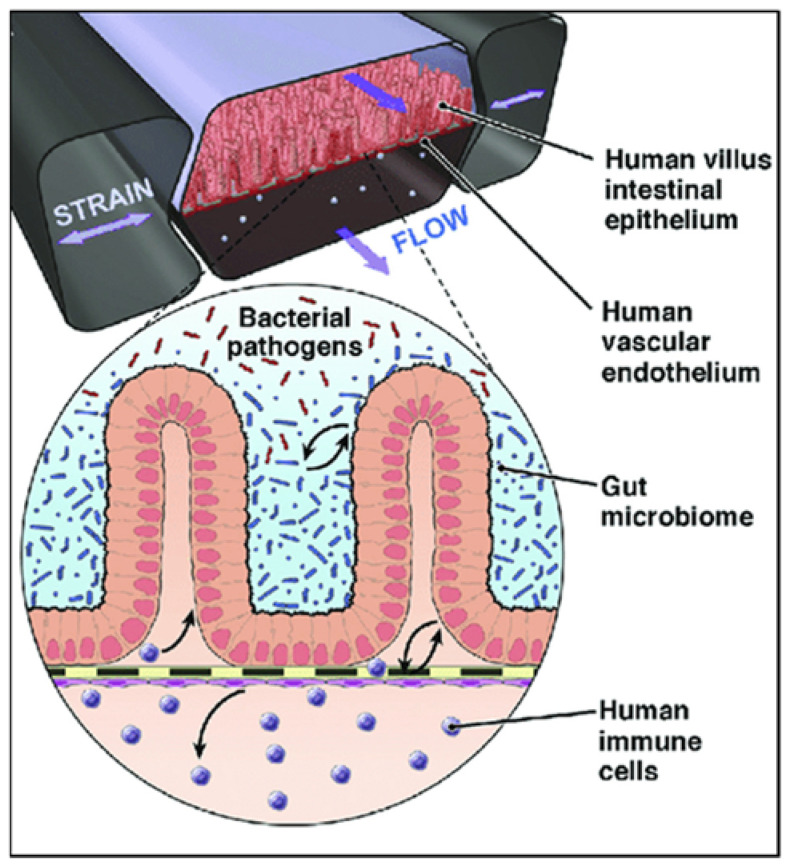
Gut-on-chip microfluidic device with human villus intestinal epithelium and human vascular endothelium formed on a flexible membrane with an active flow and peristalsis-like motion and capabilities to co-culture with bacteria and human immune cells. Adapted with permission from Bein et al. [198] (Elsevier license number: 5282361055514).

## Figures and Tables

**Figure 1 metabolites-12-00499-f001:**
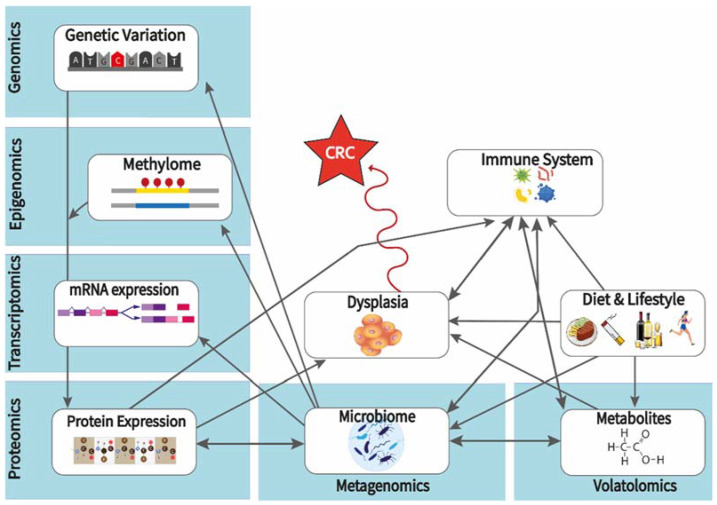
Environmental and molecular factors, biological modalities cross-linked in an overview of lifestyle, genomic, epigenomic, transcriptomic, proteomic, metabolomic, and metagenomic interactions leading to CRC-predisposing dysplasia.

**Figure 2 metabolites-12-00499-f002:**
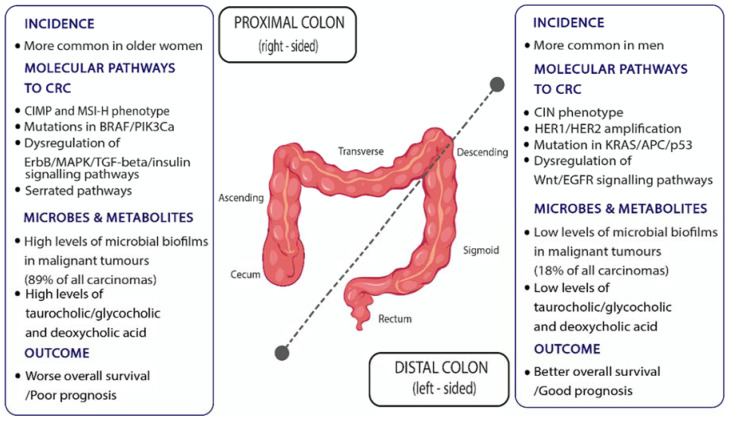
Proximal and distal colon regionalization in terms of CRC incidence, outcome, molecular pathways leading to CRC, and microbes and metabolites involved.

**Figure 3 metabolites-12-00499-f003:**
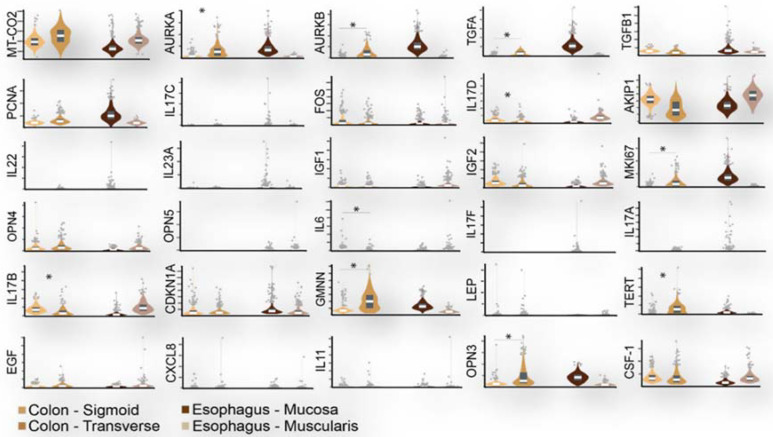
Mitosis and regenerative inflammation genes can serve as biomarkers of risk because they vary among individuals and intestinal site. Violin plots depict a wide inter-individual distribution in expression of 30 mitosis and regenerative inflammation genes in the human colon (sigmoid muscularis and transverse muscularis and mucosa) and oesophagus (mucosa and muscularis). * denote >2 or <0.5 gene expression fold change between sigmoid muscularis (*n* = 318) and transverse muscularis and mucosa (*n* = 368) calculated in Transcripts Per Million from RNA-Seq data retrieved from GTEx Analysis Release V8 (dbGaP Accession phs000424.v8.p2; https://www.gtexportal.org/home/datasets (accessed on 4 May 2022)).

**Figure 4 metabolites-12-00499-f004:**
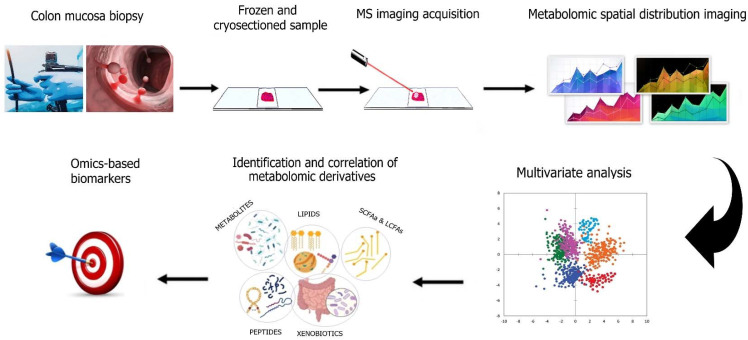
Overview of the MS imaging workflow towards CRC metabolomics. Preparation steps require the collection of healthy, adjacent-to-polyps, or tumorous colonic mucosa specimens, followed by snap freezing and cryosectioning of tissues onto compatible glass slides. Imaging requires ionization of desorbed molecules across the thin tissue surface followed by rastering. The reconstruction of metabolomic spatial distribution maps produced allows the multivariate statistical analysis of the metabolomic profile on the colonic specimen. The ensuing classification and quantification of all the metabolomic derivatives may be combined with other omic platforms to provide combinatorial, multi-omics-based biomarkers potentially applicable toward CRC prevention, diagnosis, or prognosis upon treatment.

**Table 1 metabolites-12-00499-t001:** Variations in bacteria found in faecal matter between healthy and CRC patients or patients with advanced adenomas and removed polyps, or Inflammatory Bowel Disease (IBD) patients [83,84,85,86]. [↑]: increased abundance, [↓]: decreased abundance.

		Healthy Individuals	CRC Patients	Advanced Adenoma & Removed Polyp patients	IBD Patients
**Microbiota**	**Phylum level**	*Firmicutes* ↑	*Proteobacteria* ↑		*Firmicutes* ↓
		*Actinobacteria* ↑	*Fusobacteria* ↑		*Proteobacteria* ↑
			*Lentisphaerae* ↑		
	**Genus level**		*Escherichia – Shigella* ↑		*Clostridium* ↑
			*Parvimonas* ↑		
			*Fusobacterium* ↑		
			*Porphyromonas* ↑		
			*Staphylococcus* ↑		
			*Pepto-streptococcus* ↑		
			*Peptococcus* ↑		
	**OTUs**	*Firmicutes* ↑	*Gamma-proteobacteria* ↑	*Adlercreutzia* ↑	*F. prausnitzii* ↓
		*Clostridiales* ↑	*Enterobacteriaceae* ↑		*B. adolescentis* ↓
		*Clostridia* ↑	*Fusobacteriales* ↑		*D. invisus* ↓
		*Lachnospiraceae* ↑	*Erysipelotrichaceae* ↑		*Clostridium defficil (cluster XIVa)* ↓
		*Ruminococcaceae* ↑	*Lachnospiraceae* ↓		*R. gnavus* ↑
		*Selenomonadales* ↑			*Lactobacilli* ↑
		*Negativicutes* ↑			Adherent-invasive *E. coli* ↑
		*Faecalibacterium* ↑			Adherent-invasive *Campylobacter concisus* ↑
					Enterotoxigenic *B. fragilis* (ETBF) ↑
					*B. vulgatus* ↑
					*Fusobacterium varium* ↑
					*Klebssiella pneumonie* ↑
					*Roseburia hominis* ↓
					*Faecalibacterium* ↓
					*Mycobacterium anium paratuberculosis* ↑

**Table 2 metabolites-12-00499-t002:** Comparative assessment of metabolomic techniques in terms of breadth of compounds detected, sensitivity, and spatial resolution on tissues, quantitative accuracy, type of sample material, and sample preparation required [124,127,131,132,133]. NMR: Nuclear Magnetic Resonance; GC-MS: Gas Chromatography-Mass Spectrometry; LC-MS: Liquid Chromatography-Mass Spectrometry; UHPLC-MS: Ultra High-Performance Liquid Chromatography-Mass Spectrometry; CE-MS: Capillary Electrophoresis-Mass Spectrometry; MALDI MSI: Matrix-Assisted Laser Desorption Ionization Mass Spectrometry Imaging; DESI MSI: Desorption Electrospray Ionization Mass Spectrometry Imaging; SIMS I: Secondary Ion Mass Spectroscopy Imaging; EASI MSI: Easy ambient sonic spray ionization Mass Spectrometry Imaging.

Metabolic Method	Breadth of Compounds	Detection Sensitivity	Quantitative Accuracy	Sample Material	Sample Preparation
**NMR**	Biomolecules, including metabolites	μΜ to mM	Yes	Biofluids and tissues	Minimal
**GC-MS**	Thermally stable volatiles (fatty & organic acids, steroids, di-glycerides, sugars, sugar alcohols)	<μM	Yes	Biofluids and tissues	Multiple steps/ Chemical derivatization
**LC-MS &** **UHPLC-MS**	Polar & non-polar metabolites, ribonucleotides, amino acids, amines, sugars, organic acids	pM to nM	Yes	Biofluids and tissues	Minimal
**CE-MS**	Polar metabolites (wider spectrum than LC/MS), ionic compounds	nM	Yes	Biofluids and tissues	Minimal
**MALDI MSI**	Metabolites, lipids, peptides, glycans, proteins, drugs, drug metabolites	0.5 μm to 100 μm depending on instrumentation	No	Biological tissue sections	Minimal or multi-step
**DESI MSI**	Metabolites, peptides	~50μm spatial resolution	Semi-quantitative	Biological tissue sections	No
**SIMS I &** **EASI MSI**	Metabolites, peptides	nm to mm sample surface resolution	Yes	Biological tissue sections	Minimal

**Table 3 metabolites-12-00499-t003:** Variations in metabolites found in faecal matter or ascending/descending colon biopsies between healthy and CRC patients or patients with advanced adenomas and removed polyps, or IBD patients or adults with increased BMI (overweight and obese individuals) [83,84,85,124,136,137]. [↑]: increased abundance, [↓]: decreased abundance, {asc}: ascending/right colon, {desc}: descing/left colon.

	Healthy Individuals	CRC Patients	Advanced Adenoma & Removed Polyp Patients	IBD Patients	Overweight/Obese Individuals
**Metabolites**	Sugars (maltose, fructose, iditol, glycerol, sedoheptulose) ↑	Polyamines (cadaverine, putrescine, 1,4-Butanediamine) ↑	Triacyloglycerol ↑	Methylamine, trimethylamine ↓	Trimethylamine N-oxide (TMAO) ↑ *{asc}*
Sugar alcohols ↑	Amino acids (Pro, Glu, Phe, Ala, Lys, 5-oxo-Pro, Val, Leu, Orn) ↑	2-arachidonoylglycerol ↓	SCFAs (Acetate, butyrate) ↓	Endocannabinoids (linoleoylethanolamine, oleoylethanolamine) ↓ *{asc}*
Amines (galactosamine) ↑	Cholesteryl esters (ChoE) ↑	3-phosphoglycerate ↓	Amino acids: Ala, Iso, Leu, Lys, Val ↑ *[faecal matter]*	Chenodeoxycholate ↑
Organic and fatty acids (octadecanoic acid, hexadecenoic acid, benzenepropanoic acid, linoleic acid, oleic acid) ↑	Sphingomyelin classes ↑	6-phosphoglyconate ↓	Amino acids: Ala, Cho, Glu, Iso, Leu, Val ↓ *[colon mucosa tissue]*	Cholate ↑ *[desc}*
Mannitol ↑	Glycerophosphatidylcholine ↑	1-dihomo-linoleuylglycerol ↓ *{asc}*	Amino acids: Arg, Lys ↑ *[faecal matter]*	Taurodeoxycholate ↑ *{asc}*
Poly- and monounsaturated fatty acids ↑		Aspartate ↓ *{asc}*	Taurine ↑	3-hydroxybutyrate (BHBA) ↑
Deoxycholic acid ↑		Glycerophosphorycholine (GPC) ↓ *{asc}*	Cadaverine ↑	2-arachidonoyglycerol ↑
		Glutarate ↓ *{desc}*	Indole ↑	Long chain fatty acids ↑ *{desc}*
		2-hydroxyarachidate ↓ *{desc}*	Anti-oxidants ↑	Heptadecanoic acid (margarate) ↑ *{desc}*
			Myoinositol ↑	
			Betaine ↑	
			Glycerophosphorylcholine ↑	
			Lactate, formate, glutamate ↓	
			Succinate ↓	
			Phenolic compounds ↑	
			Glycerophospoglycine ↑	
			Glucose ↑	
**Metabolic Pathways**	MAsp metabolism ↑	Asp metabolism ↑		SCFA synthesis ↓	
Ala metabolism ↑	Ammonia recycling ↑		Amino acid biosynthesis ↓	
Protein biosynthesis ↑	Protein biosynthesis ↑			
Glu-Ala cycle ↑	Trp metabolism ↑			
Selenoamino acid metabolism ↑				
Mitochondrial electron transport chain ↑				
Ammonia recycling ↑				
Glutamate metabolism ↑				
Urea cycle ↑				
Citric acid cycle ↑				
Methionine metabolism ↑				
Galactose metabolism ↑

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
