# Peer review of "Colon Cancer: From Epidemiology to Prevention"

_metabolites, 2022, doi:10.3390/metabo12060499_

Round 1

Reviewer 1 Report

The presented MS reports a current state of art concerning colorectal cancer (CRC) one of the most common types of cancer, affecting a large part of the population both in highly developed countries and in those with lower living standards. The epidemiology, the role of environment, genetic and molecular pathways, intestinal microbiota, and metabolites leading to or facilitating the 58 development of CRC are the main topics of this review report.

My impression is of a comprehensive and well-structured review that can benefit both physicians and molecular biologists.

I have only small technical remarks:

-on row 273 appears (Xia Z. et al., 2020), instead of reference [52];

-please check the correct numbering of references in this paragraph;

-reference [93] is missing in to the text;

-on the rows 438-440 The large intestine (i) mucosa and glands, (ii) the submucosa and lamina propria connective, vascular and muscle tissues, and (iii) neurons derive from endodermal, mesodermal, and neural crest embryonic tissues, respectively the sentence is uncompleted please correct it;

-in references on row 627 additional number 5 is appears and in a few other the journal are not correctly abbreviated.

Author Response

“The presented MS reports a current state of art concerning colorectal cancer (CRC) one of the most common types of cancer, affecting a large part of the population both in highly developed countries and in those with lower living standards. The epidemiology, the role of environment, genetic and molecular pathways, intestinal microbiota, and metabolites leading to or facilitating the development of CRC are the main topics of this review report.

My impression is of a comprehensive and well-structured review that can benefit both physicians and molecular biologists. I have only small technical remarks: 

“-on row 273 appears (Xia Z. et al., 2020), instead of reference [52];

-please check the correct numbering of references in this paragraph;

-reference [93] is missing in to the text

Xia Z et al 2020 is now referenced correctly in line 282. The references in that paragraph now much the text. There is also continuity in numbering around the former reference 93 (now reference 103 and 104) mentioned by the reviewer.

“-on the rows 438-440 The large intestine (i) mucosa and glands, (ii) the submucosa and lamina propria connective, vascular and muscle tissues, and (iii) neurons derive from endodermal, mesodermal, and neural crest embryonic tissues, respectively the sentence is uncompleted please correct it

This sentence is now fixed as follows (lines 480-482): “The large intestine contains: (i) the mucosa and glands, (ii) the connective tissue of submucosa and lamina propria, the vasculature and muscle, and (iii) the neurons. These derive from the embryonic endoderm, mesoderm, and neural crest, respectively.”

“-in references on row 627 additional number 5 is appears and in a few other the journal are not correctly abbreviated.

Numbering and journal abbreviation of references are now edited.

Reviewer 2 Report

Manuscript title:  Colon Cancer: from Epidemiology to Prevention

Manuscript # metabolites-1694640

General Comments

This manuscript is a review of potential causal mechanisms of colorectal carcinogenesis that aims to demonstrate the interplay of environmental and molecular factors in colorectal cancer development. While this manuscript aims to cover a comprehensive range of topics, it can be further strengthened with more consistent and complete coverage of the existing data.  A few specific comments are listed below.

Specific Comments

  1. 2.1 Modifiable risk factors with a clear environmental component: It is a bit unclear how comprehensive the authors intended to make subsections 2.1.1-2.1.7.  For example, evidence on tumor mutations are included in 2.1.2 Tobacco consumption but are omitted in 2.1.4 Alcohol consumption.
  2. 2.1.1 Inactivity, sedentary lifestyles and obesity: There have been several recent and important studies conducted on sedentary/sitting time and cancer and other outcomes, which would strengthen this section.
  3. 2.1.5 Dietary fibre and wholegrains: There is more recent evidence that explores the potential mechanism of action of dietary fiber on gut microbiota, such as the production of butyrate, that would be worth noting.
  4. 2.1.6 Dairy products and calcium supplements: Although the authors’ intent is understood, vitamin D3 is not a milk derivative.
  5. 2.1.6 Self-reported Race/Ethnicity: A more specific link is needed for ref #39, rather than the general website for the American Cancer Society.
  6. 5. Intestinal metabolomics as an emerging way to study CRC prevention: The use of the term “shed” is an interesting choice and could be somewhat misleading.

Author Response

“General Comments

This manuscript is a review of potential causal mechanisms of colorectal carcinogenesis that aims to demonstrate the interplay of environmental and molecular factors in colorectal cancer development. While this manuscript aims to cover a comprehensive range of topics, it can be further strengthened with more consistent and complete coverage of the existing data.  A few specific comments are listed below.

Specific Comments

2.1 Modifiable risk factors with a clear environmental component: It is a bit unclear how comprehensive the authors intended to make subsections 2.1.1-2.1.7.  For example, evidence on tumor mutations are included in 2.1.2 Tobacco consumption but are omitted in 2.1.4 Alcohol consumption.

We added evidence describing tumor mutations in the section 2.1.4 in lines 173-175.

“2.1.1 Inactivity, sedentary lifestyles and obesity: There have been several recent and important studies conducted on sedentary/sitting time and cancer and other outcomes, which would strengthen this section.

We extensively revised this section (lines 103-121) to include recent studies.

“2.1.5 Dietary fibre and wholegrains: There is more recent evidence that explores the potential mechanism of action of dietary fiber on gut microbiota, such as the production of butyrate, that would be worth noting.

We extensively revised this section (lines 178-188) to include studies on SCFAs.

“2.1.6 Dairy products and calcium supplements: Although the authors’ intent is understood, vitamin D3 is not a milk derivative.

To clarify we renamed section 2.1.6 to “Dairy products and dietary supplements” and edited current line 195 to mention that Vitamin D3 is the main facilitator of calcium and phosphorus absorption in the body.

“2.1.6 Self-reported Race/Ethnicity: A more specific link is needed for ref #39, rather than the general website for the American Cancer Society.

The reviewer meant to say section 3.1.6. In this section references 67 and 68 are now added.

“5. Intestinal metabolomics as an emerging way to study CRC prevention: The use of the term “shed” is an interesting choice and could be somewhat misleading.

To clarify the pertinent line in the text is now revised to read (lines 412-413): “Humans take up dietary nutrients and work along with the gut microbiomes to produce metabolites”.